# Quercetin Inhibits Cell Survival and Metastatic Ability via the EMT-Mediated Pathway in Oral Squamous Cell Carcinoma

**DOI:** 10.3390/molecules25030757

**Published:** 2020-02-10

**Authors:** So Ra Kim, Eun Young Lee, Da Jeong Kim, Hye Jung Kim, Hae Ryoun Park

**Affiliations:** 1Department of Oral Pathology, and BK21 PLUS Project, School of Dentistry, Pusan National University, Yangsan 50612, Korea; ksr9307@pusan.ac.kr (S.R.K.); eunyeong26@pusan.ac.kr (E.Y.L.); ttakjjung@gmail.com (D.J.K.); 2Periodontal Disease Signaling Network Research Center (MRC), School of Dentistry, Pusan National University, Yangsan 50612, Korea

**Keywords:** quercetin, oral squamous cell carcinoma cells, metastasis, cell cycle arrest, epithelial-to-mesenchymal transition, matrix metalloproteinase, transforming growth factor-β1

## Abstract

This study aimed to investigate whether quercetin exerts anticancer effects on oral squamous cell carcinoma (OSCC) cell lines and to elucidate its mechanism of action. These anticancer effects in OSCC cells were assessed using an MTT assay, flow cytometry (to assess the cell cycle), wound-healing assay, invasion assay, Western blot analysis, gelatin zymography, and immunofluorescence. To investigate whether quercetin also inhibits transforming growth factor β1 (TGF-β1)-induced epithelial–mesenchymal transition (EMT) in human keratinocyte cells, HaCaT cells were treated with TGF-β1. Overall, our results strongly suggest that quercetin suppressed the viability of OSCC cells by inducing cell cycle arrest at the G2/M phase. However, quercetin did not affect cell viability of human keratinocytes such as HaCaT (immortal keratinocyte) and nHOK (primary normal human oral keratinocyte) cells. Additionally, quercetin suppresses cell migration through EMT and matrix metalloproteinase (MMP) in OSCC cells and decreases TGF-β1-induced EMT in HaCaT cells. In conclusion, this study is the first, to our knowledge, to demonstrate that quercetin can inhibit the survival and metastatic ability of OSCC cells via the EMT-mediated pathway, specifically Slug. Quercetin may thus provide a novel pharmacological approach for the treatment of OSCCs.

## 1. Introduction

Oral squamous cell carcinoma (OSCC) is the most common malignancy, accounting for about 90% of head and neck cancers, and develops at the lips, tongue, salivary glands, gums, bottom of the mouth, pharynx, and surfaces within the oral cavity [1,2,3,4,5,6]. OSCC, an aggressive human cancer with the highest mortality, has a significant impact on the quality of life of patients [6,7]. Early OSCC is treated with a combination of surgery and radiation therapy; however, this strategy does not significantly increase survival in OSCC patients. Thus, many studies have focused on compounds derived from natural products as potential candidate agents for treating or preventing OSCCs. OSCC is known to have multifactorial etiologies, including tobacco, alcohol, and viruses [6,8,9,10]. Several reviews suggest that periodontal disease (PD) results in a high risk of developing OSCC [8,11]. In ”Cancer Facts & Figures 2018” provided by the American Cancer Society, the oral cavity is the most common site of onset of oral cancer, followed by the tongue and pharynx [12]. Therefore, we used oral squamous cell carcinoma HN22 and tongue squamous cell carcinoma such as OSC20 and SAS to confirm the effect of quercetin in oral cancer.

Cancer metastasis involves a complex process that includes primary tumor formation, angiogenesis, invasion, and metastatic colonization [8]. In the initial stages of cancer metastasis, epithelial–to–mesenchymal transition (EMT) causes a loss of adhesion and increases the motility of primary tumor cells [3,13,14,15]. The matrix metalloproteinase (MMP) is an important extracellular protease that breaks down the extracellular matrix (ECM) and allows cancer cells to migrate into the lymph and blood vessels [16,17,18]. Some studies have demonstrated that EMT and MMP support the invasion and metastasis of OSCC [19,20]. Therefore, this study investigated whether quercetin exhibits anticancer effects on OSCC cell lines and its mechanism of action.

Quercetin is a plant flavonol of the flavonoid polyphenol group found in many fruits, vegetables, leaves, and grains [21,22,23]. Although numerous studies have demonstrated that quercetin plays a role in the prevention of various cancers [24,25,26], it is still an interesting target for OSCCs. Previous studies have reported the antibacterial effects of quercetin against *Actinobacillus actinomycetemcomitans (Aa)* and *Porphyromonas gingivalis (Pg)*, which are the major pathogens in PD [27,28]. Moreover, recently, studies have reported that quercetin inhibits the cell viability, migration, and invasion of OSCC cells [29,30,31,32]. However, its potential actions in OSCCs have been poorly explored. Therefore, in this study, we sought to investigate whether quercetin exhibits anticancer effects on OSCC cell lines and to elucidate its mechanism of action.

## 2. Results

### 2.1. Quercetin Reduced Cell Viability and Arrested G2 Phase Cell Cycle in OSCC Cells

To determine the effect of quercetin on cell viability, OSCC cell lines (OSC20, SAS, and HN22 cells) were treated with various concentrations (0~160 µM) of quercetin for 24 h, and cell viability was determined using the MTT assay. The results showed that quercetin decreased cell viability in OSCC cell lines (Figure 1A). In particular, cell viability in OSC20 and SAS cells, except HN22 cells, was reduced in a quercetin dose-dependent manner. Treatment with 160 μM quercetin for 24 h reduced cell viability by approximately 50% in all cell lines (*p* < 0.01), whereas treatment with low concentrations (10–40 μM) reduced cell viability slightly (approximately 5–30%) (*p* < 0.05) compared to the control (Figure 1A). To determine the function of quercetin in the regulation of the cell cycle, OSCC cell lines were stimulated with low levels of quercetin (10–40 μM) for 24 h and analyzed the distribution of the cell cycle using flow cytometry. As shown in Figure 1B, the cell populations at the G2/M stage were increased by treatment with quercetin in the OSCC cell lines (OSC20 cell, 7.66% ± 3.92% in the control vs. 36.22% ± 4.10% in quercetin 40 µM, *p* < 0.01; SAS cell, 22.85% ± 3.04% in the control vs. 45.49% ± 1.38% in quercetin 40 µM, *p* < 0.005; and HN22 cell, 23.37% ± 4.13% in control vs. 50.40% ± 5.05% in quercetin 40 µM, *p* < 0.05). However, the cell proportion in the G0/G1 phase decreased (OSC20 cell, 35.99 ± 1.89% in the control vs. 14.06% ± 1.26% in quercetin 40 µM, *p* < 0.001; SAS cell, 33.03% ± 1.12% in the control vs. 8.14% ± 4.39% in quercetin 40 µM, *p* < 0.005; and HN22 cell, 35.05% ± 1.62% in control vs. 6.53% ± 4.34% in quercetin 40 µM, *p* < 0.001). The results indicate that quercetin suppressed the viability of OSCC cells by inducing cell cycle arrest in the G2/M phase.

### 2.2. Quercetin Suppressed the Migration Potential of OSCC Cells

We performed a wound-healing assay to evaluate the inhibitory effects of quercetin on migration. To observe the effects of quercetin on cell migration, the cell proliferation of OSC20, SAS, and HN22 cells was inhibited by treatment with thymidine (1 mM) for 2 h prior to treatment with quercetin (40 μM). After 24 h, the wound area of the control was almost completely reduced compared to the initial area; however, the quercetin-treated cells did not show any decrease (Figure 2A,B). The migration of quercetin-treated cells was significantly decreased in a dose-dependent manner (not shown). This result suggests that the migratory properties were completely lost upon quercetin treatment for 24 h in the OSCC cell lines.

### 2.3. Quercetin Regulated EMT and MMPs in OSCC Cells

The effects of quercetin treatment on EMT and the activity of ECM-degrading enzymes in OSCC cell lines were analyzed by Western blot or gelatin zymography. OSC20, SAS, and HN22 cells were treated with different concentrations of quercetin (0, 10, 20, and 40 μM) for 24 h. As shown in Figure 3A, quercetin, in a dose-dependent manner, increased the expression of epithelial markers, such as E-cadherin and claudin-1, while decreasing the expression of mesenchymal markers, such as fibronectin, vimentin, and alpha-smooth muscle actin (α-SMA). We observed that 40 μM of quercetin specifically downregulated the expression of the mesenchymal markers in OSCC cells. The activation of MMP-2 and MMP-9 was significantly decreased by quercetin treatment (Figure 3B). These results demonstrate that quercetin suppressed the expression levels of EMT inducers and MMPs in OSCC cells.

### 2.4. Quercetin Regulated EMT-Activating Transcription Factors in OSCC Cells

EMT-activating transcription factors (EMT-TFs), such as Twist, Slug, and Snail 1, were observed by Western blot and immunofluorescence. Twist and Slug were significantly downregulated in quercetin-treated OSC20 and SAS cells compared to the untreated cells (Figure 4A,B). Slug was significantly downregulated in quercetin-treated HN22 cells compared to untreated cells, but Twist was unchanged (Figure 4A,B). Moreover, upon quercetin treatment, translocation of Twist and Slug expression was observed in the OSC20 and SAS cells by immunofluorescence (Figure 4C,D). The basal level of Twist was observed in the cytosol, and there was no translocation upon quercetin treatment in HN22 cells (Figure 4C). The basal levels of Slug were observed in the nucleus on untreated HN22 cells but were observed in the cytosol upon quercetin treatment (Figure 4D). These data strongly suggest that quercetin inhibits Slug activation by inducing its translocation into the cytoplasm.

### 2.5. Transforming Growth Factor β1 (TGF-β1) Induced EMT in Human Keratinocyte HaCaT Cells

Previous results indicated that quercetin exerts anticancer effects through the regulation of EMT in OSCC cells. To examine the viability effect of quercetin on human keratinocyte cells, HaCaT (immortal keratinocyte) and nHOK (primary normal human oral keratinocyte) cells were treated with various concentrations (0~160 µM) of quercetin for 24 h, and cell viability was determined by an MTT assay. Cell viability of HaCaT cells and nHOK cells was not changed by the low concentration (10–40 μM) of quercetin compared to the control group but was significantly reduced in 160 μM quercetin treatment (*p* < 0.05) (Figure 5A). The results show that quercetin had not much effect on cell viability of HaCaT and nHOK cells in the concentration range of 10–80 μM compared to the OSCC cell lines. To investigate whether quercetin also inhibits TGF-β1-induced EMT in human keratinocyte, HaCaT cells were treated with TGF-β1 (2 ng, 5 ng, and 10 ng) for 24 h. Changes in the expression of EMT inducers upon treatment with TGF-β1 were confirmed by Western blot (Figure 5B). The expression levels of the epithelial markers (E-cadherin and claudin-1) were decreased and those of the mesenchymal markers (fibronectin, vimentin, α-SMA, and Slug) were increased (Figure 5C–H). TGF-β1 (10 ng/mL) was determined to be the optimal concentration for the EMT induction in HaCaT cells. Thus, 10 ng/mL of TGF-β1 was used for the subsequent experiment.

### 2.6. Quercetin Attenuated TGF-β1-induced EMT in HaCaT Cells

To investigate whether quercetin also inhibits TGF-β1-induced EMT in normal cells, HaCaT was treated with 10 ng of TGF-β1 for 24 h, followed by various concentrations of quercetin for 24 h without changing the media (Figure 6A). To observe the morphological changes of the treated HaCaT cells, phalloidin staining was conducted. Without treatment, the cells tended to grow together and exhibited a triangle–square shape. However, after treatment with 10 ng of TGF-β1, the HaCaT cells showed morphological changes to their spindle shapes and were spread out (Figure 6B). Additionally, when the cells were treated with TGF-β1 and quercetin together, as explained previously, the morphological changes recovered but the cells did not gather as before (Figure 6B). Western blot data show that quercetin regulated the TGF-β1-induced EMT markers. With regard to the epithelial markers (E-cadherin and claudin-1) of the induced EMT, there were no changes in recovery. In contrast, changes in the mesenchymal markers indicate that treatment with 40 μM quercetin leads to the greatest recovery. Thus, we determined the optimal concentration of quercetin as 40 μM for use in the subsequent experiment (Figure 6C–I). To evaluate the TGF-β1-induced EMT migration ability, a wound-healing assay was performed. The wound area was narrower in the TGF-β1-treated HaCaT cells than in the untreated HaCaT cells (Figure 6J,K). Further, quercetin seemed to inhibit the invasion of EMT-induced HaCaT cells (Figure 6L). The invasion assay also showed that the invasion ability improved upon TGF-β1 treatment. The number of cells that passed the Matrigel and membrane was less in the HaCaT cells treated with quercetin and TGF-β1 than in those treated with only TGF-β1 (Figure 6L). In conclusion, quercetin decreased the TGF-β1-induced EMT in HaCaT cells.

## 3. Discussion

Suppression of cancer metastasis is considered an effective strategy for preventing cancer progression. Many studies have reported that a variety of natural products and compounds exert antitumor effects by inducing apoptosis and cell cycle arrest in cancer cells and reducing their metastatic capacities [33,34]. Thus, the use of dietary biologically active compounds may be a safe and desirable approach for cancer treatment. Quercetin is a potent candidate for cancer treatment, because it is known to have a regulatory effect on apoptosis, migration, and invasion. However, quercetin has limited applications as a therapeutic agent due to problems such as low oral bioavailability (8~16%) and poor water solubility [35]. Numerous studies have used various cancer cell lines to study the anticancer effects of quercetin, and quercetin concentrations vary depending on the cell line and treatment time. Quercetin is used at 10–80 μM concentrations for 24 treatments in cancer cells because of its low bioactivity. Quercetin concentrations 10–80 μM are used under 24-h treatment conditions. Quercetin IC50 values of quercetin are 60–120 μM [36,37,38]. Especially, quercetin of 25–50 μM had anticancer effects in SAS cells, a typical oral cancer cell [39]. In this study, the anticancer effect of quercetin at 10, 20, 40, 80, and 160 μM was assessed in vitro by MTT assay in the OSCC cell line, such as OSC20, SAS, and HN22 cells. The optimal concentration of quercetin was determined to be 10–40 μM for assessing metastatic capacity in OSCC cells. The 10–40 μM quercetin concentration used in this study is commonly used for cancer cells under 24-h treatment conditions. Quercetin, a flavonoid found in many fruits, vegetables, and grains, has been shown to have an antitumor effect on OSCC cells via the induction of cell cycle arrest [29,40]. Quercetin treatment suppressed cell growth by inducing G2/M arrest and apoptosis in EGFR-overexpressing HSC-3 and TW206 oral cancer cells [30]. Our result show that quercetin inhibits the viability of OSCC cell lines, such as OSC20, SAS, and HN22 cells, in a dose-dependent manner by inducing cell cycle arrest at the G2/M phase (Figure 1A,B).

The EMT process contributes to greater interstitial cell adhesion profiles, as unregulated cellular or cell matrix adhesion and increased migration and invasion are required for the development of metastatic cancer [41]. The ability of cancer cells to migrate and invade is increased by the reduction of the expression of E-cadherin, a cell–cell adhesion molecule, whereas the expression of N-cadherin is decreased, which is inversely proportional to that of E-cadherin [42]. Quercetin regulates the expression of EMT markers, such as E-cadherin, vimentin, and β-catenin, in head and neck cancer-derived sphere cells and breast cancer cells [43,44]. In addition, quercetin prevented the EGF-induced expression of N-cadherin and vimentin and increased the expression of E-cadherin in prostate cells [45]. Based on these reports, it is assumed that quercetin treatment can modulate the expression of EMT-related markers. Additionally, quercetin suppressed cell viability, migration, and invasion by regulating miR-16/HOXA10, which inhibits the expression of MMP-2 and 9 in oral cancer cells [30]. Studies on the mechanism of action of quercetin in the metastasis of OSCCs are still insufficient. In this study, quercetin inhibited EMT by increasing the expression of E-cadherin and claudin-1 and decreasing the expression of fibronectin, vimentin, α-SMA, and Slug in OSCC cell lines (Figure 3 and Figure 4). The activation of MMP-2 and MMP-9 was significantly decreased by quercetin treatment (Figure 3B). Thus, quercetin inhibits cell migration and invasion by regulating the expression of EMT-related genes in OSCC cell lines.

It has been found that transforming growth factor-beta 1 (TGF-β1) induces epithelial mesenchymal transition (EMT) in various cancer cells, promoting motility and invasiveness [46,47]. Additionally, TGF-β1 treatment reduces the enhanced expression of Snail1 and Twist1 and, subsequently, the expression of E-cadherin [47]. We observed whether quercetin reduced EMT-related factors increased by TGF- β1 treatment in human keratinocyte HaCaT cells. TGF-β1 can promote tumor formation by inducing mesenchymal metastasis (via EMT) in the epithelium in various cancer cells. The main TGF-β1-induced effectors in this process are EMT transcription inhibitors, such as Snail 1, Snail 2 (also known as Slug), ZEB1/2, and Twist. HaCaT cells showed morphological changes and induction of EMT-related markers upon TGF-β1 treatment (Figure 6B,C). Quercetin reduced TGF-β1-induced EMT expression in HaCaT cells.

Overall, our results strongly suggest that quercetin suppressed the viability of OSCC cells inducing cell cycle arrest in the G2/M phase (Figure 1). Significantly, our results demonstrate that quercetin suppresses the metastatic ability via EMT and MMP in OSCC cells and decreases the expression level of TGF-β1-induced EMT in human normal HaCaT cells. Thus, quercetin may provide a novel pharmacological approach for the treatment of OSCCs.

## 4. Materials and Methods

### 4.1. Reagents and Antibodies

Quercetin was purchased from Sigma-Aldrich (St. Louis, MO) and dissolved in DMSO (DUKSAN, Korea) to prepare stock solutions (100 mM). TGF-β1 was purchased from R&D Systems (Minneapolis, USA) and dissolved in 4 mM HCl containing 0.1% fatty acid-free bovine serum albumin to a concentration of 1.0 mg/mL to prepare stock solutions (20 μg/mL). 3-(4,5-dimethylthiazol-2-yl)-2,5-diphenyltetrazolium bromide (MTT) and thymidine were purchased from Sigma-Aldrich (St. Louis, MO). Antibodies against E-cadherin, vimentin, and Slug were purchased from Cell Signaling Technology (Danvers, MA, USA), whereas fibronectin was purchased from BD Bioscience (San Jose, CA, USA); α-smooth muscle actin was purchased from Abcam (Cambridge, MA, USA), and Twist and beta-actin were purchased from Santa Cruz Biotechnology (Santa Cruz, CA, USA).

### 4.2. Cell Culture and Treatment

Three OSCC cell lines (OSC20, SAS, and HN22 cells) and human keratinocyte cells such as HaCaT (immortal keratinocyte) and nHOK (primary normal human oral keratinocyte) were used in this study. OSC20 and SAS cells were grown in Dulbecco’s modified Eagle’s medium F-12, 1:1 mixture (Hyclone, UT, USA); HN22 and HaCaT cells were grown in Dulbecco’s modified Eagle’s medium with high glucose (DMEM; Hyclone); and nHOK (primary normal human oral keratinocyte) were cultured in KBM^TM^ Gold Keratinocyte growth basal medium (Lonza, MD, USA, #CC-3103) containing KGM^TM^-2 SingleQuots^TM^ supplements (Lonza, MD, USA, #CC-4152) at 37 °C in 5% CO_2_ and 95% O_2_ in a humidified environment. Further, both media contained 10% fetal bovine serum (FBS; Hyclone) and 1% penicillin and streptomycin (Thermo Fisher, Waltham, MA, USA). In this study, the three OSCC cell lines were treated with various concentrations of quercetin for 24 h; the HaCaT cells were treated with 10 ng of TGF-β1 for 24 h and then treated with various concentrations of quercetin.

### 4.3. MTT Assay

An MTT (3-(4,5-dimethylthiazol-2-yl)-2,5-diphenyltetrazolium bromide) assay was conducted to evaluate the cytotoxicity of quercetin on cell viability. Each cell line was seeded at 1 × 10^4^ cells per well in a 96-well plate and incubated with 10% FBS-containing media. However, the exception was that nHOK cells were seeded with 3 × 10^4^ cells per well with serum-free media. After stabilization, the media was replaced with a media containing 10% FBS for 24 h. Then, the cells were treated with various concentrations of quercetin (10, 20, 40, 80, and 160 μM) with media containing 10% FBS. After 24 h, the media were replaced with MTT (500 μg/mL) in a fresh serum-free media, and the cells were incubated for 3 h at 37 °C. Then, the supernatant was removed, and 200 μL of dimethylsulfoxide (DMSO) was added to dissolve the MTT formazan. The absorbance of the plate was read at 570 nm using a Synergy HT micro plate reader (Bio-tek, Winooski, VT, USA).

### 4.4. Cell Cycle Analysis

A cell cycle analysis was conducted to observe the cell cycle’s progression. Cells were seeded at a density of 5 × 10^5^ cells in 60 mm dishes. After cell stabilization, the cells were treated with 40 μM of quercetin for 24 h. Then, the cells were detached using trypsin and washed with phosphate-buffered saline (PBS). The harvested cells were then fixed with 75% ethanol at 4 °C overnight; subsequently, they were centrifuged and incubated with 20 μL RNase A (10 mg/mL) in 1 mL PBS at 4 °C for 30 min. Lastly, cells were stained with 25 μg/mL PI at room temperature for 10 min in the dark. The DNA content was then analyzed using a BD FACSCanto II flow cytometer (BD Bioscience, San Joes, CA, USA).

### 4.5. Wound-Healing Assay

A wound-healing assay was conducted to evaluate cell migration ability. Cells were seeded in 6-well plates at 5 × 10^5^ per well. At 80% confluence, the cells were treated with 1 mM thymidine for 2 h before reagent treatment. Then, a wound was made in the middle of the well using a sterile 200 μL pipette tip. Further, wells were washed three times with PBS. OSCC cell lines were incubated with media containing 1% FBS and 40 μM of quercetin for 24 h. HaCaT cells were incubated with media containing 1% FBS and 10 ng TGF-β1 for 24 h; then, 40 μM of quercetin was added continuously for 24 h without discarding the existing media. Images were captured using a microscope (Nikon Eclipse TS100, Japan; X100). The rate of reduction of the wound area was calculated as a percentage of the remaining wound area compared to the initial wound area.

### 4.6. Western Blot Analysis

After treatment, cells were washed three times with cold PBS and lysed with 1X RIPA buffer containing a protease inhibitor cocktail and phosphatase inhibitors. The lysed cells were centrifuged at 12,000 ×g for 24 h. Subsequently, the supernatant was collected. Using the method of Bradford, the protein concentration was quantified as 20 μg. Samples were separated on 10% SDS-polyacrylamide gels. Then, SDS-gels were transferred electrophoretically onto polyvinylidene fluoride membranes (Bio-Rad) using a wet transfer kit. The transferred membranes were blocked with 5% skim milk in Tris-buffered saline containing 0.1% Tween-20 for 1 h at room temperature. The membranes were incubated overnight at 4 °C with the primary antibodies against E-cadherin (cell signaling, #3195, 1:1000); claudin-1 (cell signaling, #4933, 1:1000); fibronectin (BD Bioscience, #610077, 1:1000); vimentin (cell signaling, #5741, 1:1000); α-SMA (Abcam, #ab5694, 1:000); slug (cell signaling, #9585, 1:1000); and Twist (Santa Cruz, #SC-81417, 1:500) and β-actin (Santa Cruz, #SC-47778, 1:4000). Continuously, the membranes were washed three times with 1X TNE buffer, and HRP-conjugated secondary antibodies (ENZO, #ADI-SAB-300-J, and #ADI-SAB-100-J, 1:8000) were applied for 3 h at room temperature. The antigen–antibody complexes were detected using a SuperSignal West–Femto reagent (Thermo Fisher, Waltham, MA, USA).

### 4.7. Gelatin Zymography

Gelatin zymography was conducted to detect matrix metalloproteinase (MMP) activity using the conditioned media. Each cell line was seeded at 5 × 10^5^ cells per well in a 6-well plate with 10% FBS-containing media. After cell stabilization, the media were transformed into serum-free media and treated with different concentrations of quercetin for 24 h at 37 °C. Subsequently, the supernatant was collected and mixed with a sample buffer (50% 0.5 M Tris-HCl (pH 6.8), 40% glycerol, 8% SDS, and 0.01% bromophenol blue). Further, samples were loaded onto a 7.5% SDS-polyacrylamide gel containing 0.2% gelatin (Sigma). The gels were washed twice with PBS containing 2.5% Triton X-100 for 1 h and incubated with an incubation buffer at 37 °C for 48 h. Then, the gels were stained with Coomassie Blue R-250 (LPS solution, Dae Jeon, Korea) for 1 h. Progressively, the gels were destained with a destaining buffer (8% acetic acid and 4% methanol) until a clear zone was visible.

### 4.8. Immunofluorescence Staining

Cells were seeded at 2 × 10^5^ cells onto a coverslip in a 6-well plate. In the case of OSCC, the cells were stained for transcription factor expression. The treated cells were fixed in 4% paraformaldehyde at room temperature for 15 min and washed three times with PBS for 10 min. Then, the cells were permeabilized by 0.02% Triton X-100 in PBS at room temperature for 20 min and washed with PBS. Diluted 1:100 primary antibodies (Slug and Twist) were applied to the cells overnight at 4 °C. On the following day, cells were washed three times with PBS, treated with fluorescein isothiocyanate (FITC)-conjugated secondary antibodies for 1 h at 37 °C, and washed three times. Then, 4 μg/mL Hoechst was used to stain the nuclei for 30 min at 37 °C. The treated HaCaT cells were stained to assess changes in their morphology. HaCaT cells were incubated with a phalloidin-conjugate working solution (1:100) and 4 μg/mL Hoechst instead of antibodies. Subsequently, the cells were washed and mounted on a slide glass. An image was then captured via confocal microscopy (LSM780, Carl Zeiss, Jena, Germany).

### 4.9. Invasion Assay

An 8 μM pore size Transwell (Corning Inc, Corning, NY, USA) was coated with 40 μL of Matrigel (BD Bioscience, San Joes, CA, USA), mixing the serum-free media at a ratio of 1:3 and air drying for 3 h at 37 °C. HaCaT cells were incubated for 24 h with or without 10 ng TGF-β1 and detached using trypsin. The cells were seeded in a Matrigel-coated Transwell culture chamber at 5 × 10^4^ cells with serum-free media containing 10 ng TGF-β1, co-treated with TGF-β1 and quercetin (10 ng TGF-β1 and 40 μM) or without treatment. Then, 600 μL of culture medium containing 10% FBS was placed onto the lower chamber. The cells were next incubated at 37 °C in a 5% CO_2_ atmosphere for 44 h. After incubation, the cells on the upper chamber were removed using a cotton swab. The cells passing through the Matrigel and membrane were fixed with cold methanol for 30 min and washed with PBS. The membrane was stained with Mayer’s hematoxylin (Muto, Tokyo, Japan) and 1% Eosin Y solution (Muto). Photographs were captured under microscopy (Olympus BX51; 40X).

### 4.10. Statistical Analyses

Statistical analyses were performed by a one-way analysis of variance (ANOVA) followed by a Bonferroni post-test, whereby three or more experimental groups were compared. *p*-values of < 0.05 were considered statistically significant. All data are the means ± standard error of the mean (SEM).

## Figures and Tables

**Figure 1 molecules-25-00757-f001:**
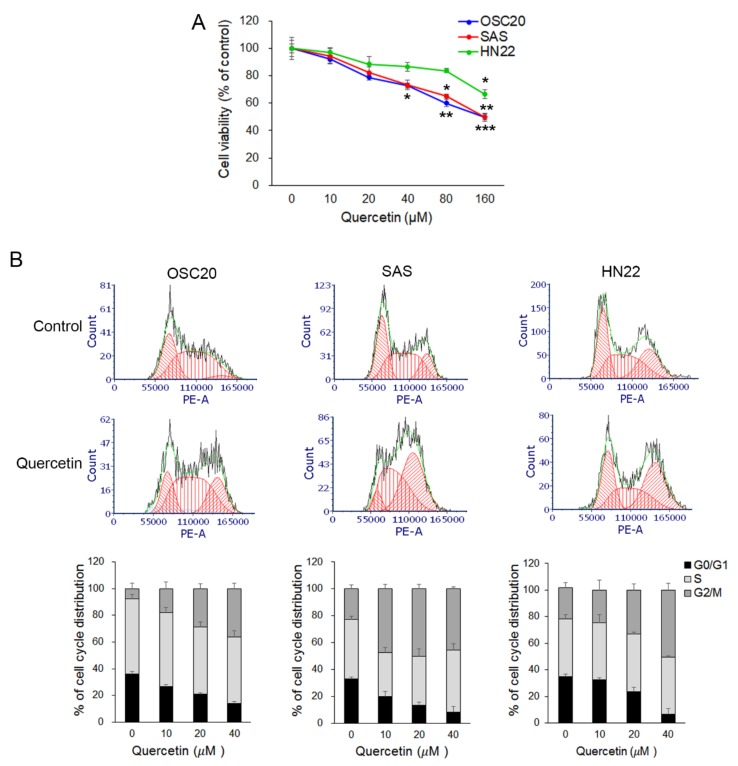
Quercetin reduced cell viability and arrested the G2/M phase cell cycle in oral squamous cell carcinoma (OSCC) cells. (**A**) Cell viability was investigated by an MTT assay. Oral squamous cell carcinoma cell lines (OSC20, SAS, and HN22 cells) were treated with quercetin (10, 20, 40, 80, and 160 μM). (**B**) Quercetin was shown to induce cell cycle arrest in OSC20, SAS, and HN22 cells. Data are the means ± SEM. * *p* < 0.05 and ** *p* < 0.01 vs. corresponding control (quercetin 0 μM).

**Figure 2 molecules-25-00757-f002:**
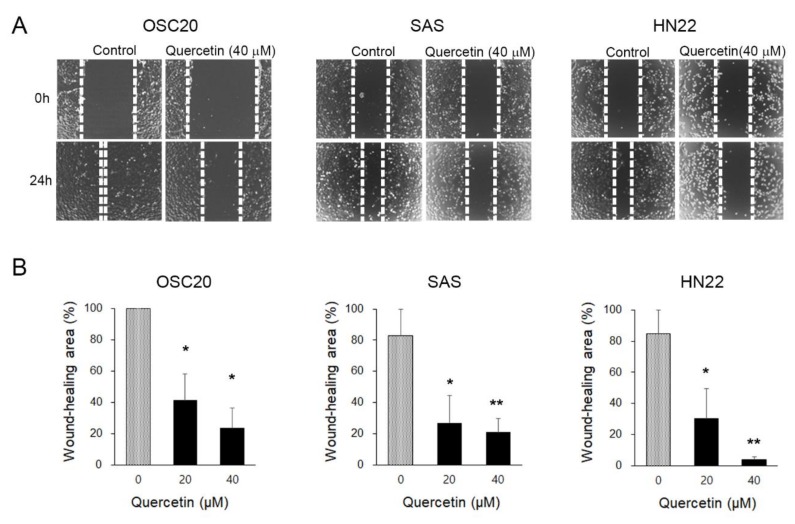
Cell migration ability assessed by a wound-healing assay. (**A**) Changes in the wound area were observed after 24 h. In the quercetin-treated cells, the wound area was less closed. This indicates a decrease in migration capacity. (**B**) The wound area was calculated and presented as a graph. Data are the means ± SEM. * *p* < 0.05 and ** *p* < 0.01 vs. the corresponding control (quercetin 0 μM).

**Figure 3 molecules-25-00757-f003:**
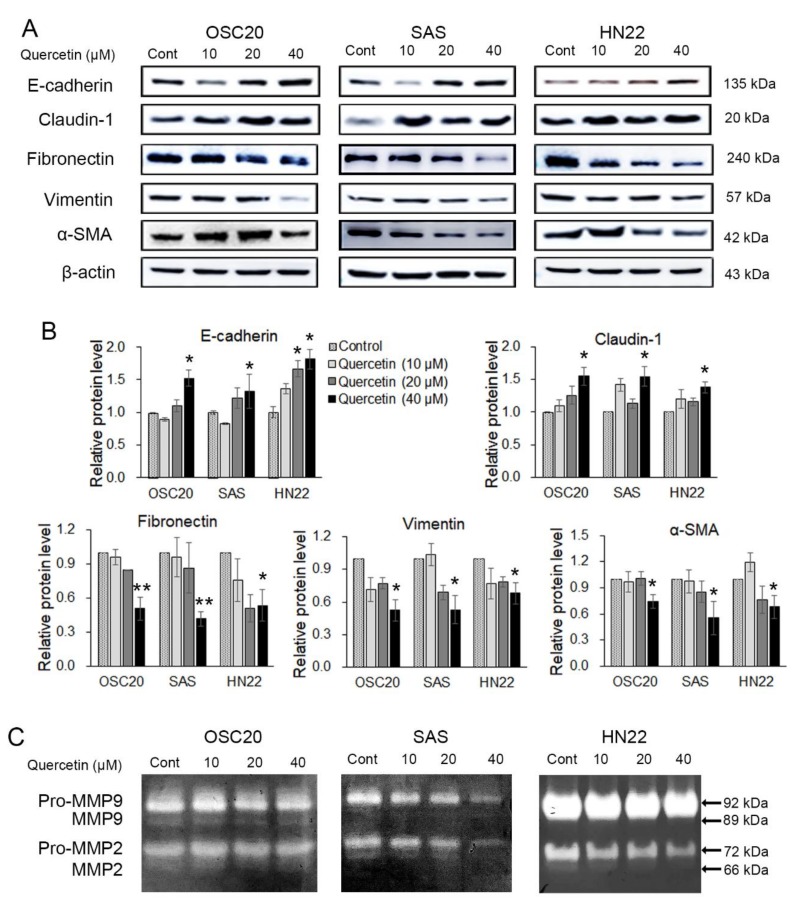
Quercetin is shown to induce regulation of epithelial-mesenchymal transition (EMT) and matrix metalloproteinase (MMP). (**A**) Western blotting was conducted to examine the changes in the EMT inducers. The results showed that the epithelial markers (E-cadherin and claudin-1) were upregulated, and the mesenchymal markers (fibronectin, vimentin, and alpha-smooth muscle actin (α-SMA)) were downregulated upon treatment with quercetin. (**B**) Quantitation of A. The band intensities of each target protein were measured using an image analyzer and presented as relative ratio. (**C**) Gelatin zymography shows the MMP-2 and MMP-9 activities in oral cancer cell lines (OSC20, SAS, and HN22 cells) upon quercetin treatment. Data are the means ± SEM. * *p* < 0.05, and ** *p* < 0.01 vs. the corresponding control (quercetin 0 μM).

**Figure 4 molecules-25-00757-f004:**
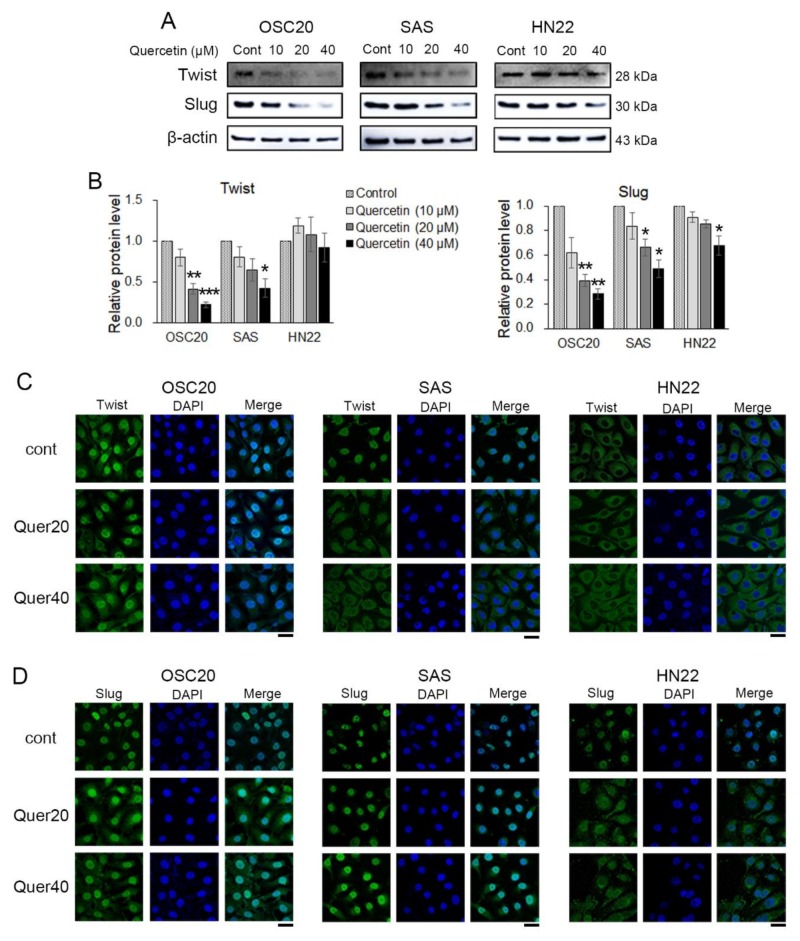
EMT-activating transcription factors were detected using Western blot and immunofluorescence. (**A**) The Western blot showed that quercetin downregulated EMT transcription factors at the protein level. (**B**) Quantitation of A. The band intensities of each target protein were measured using an image analyzer and presented as relative ratio. (**C**) Representative fluorescence microscopy images of Twist in OSCC cell lines. (**D**) Representative fluorescence microscopy images of Slug in OSCC cell lines. Scale bars: 20 μM. Data are the means ± SEM. * *p* < 0.05, ** *p* < 0.01, and *** *p* < 0.001 vs. the corresponding control (quercetin 0 μM).

**Figure 5 molecules-25-00757-f005:**
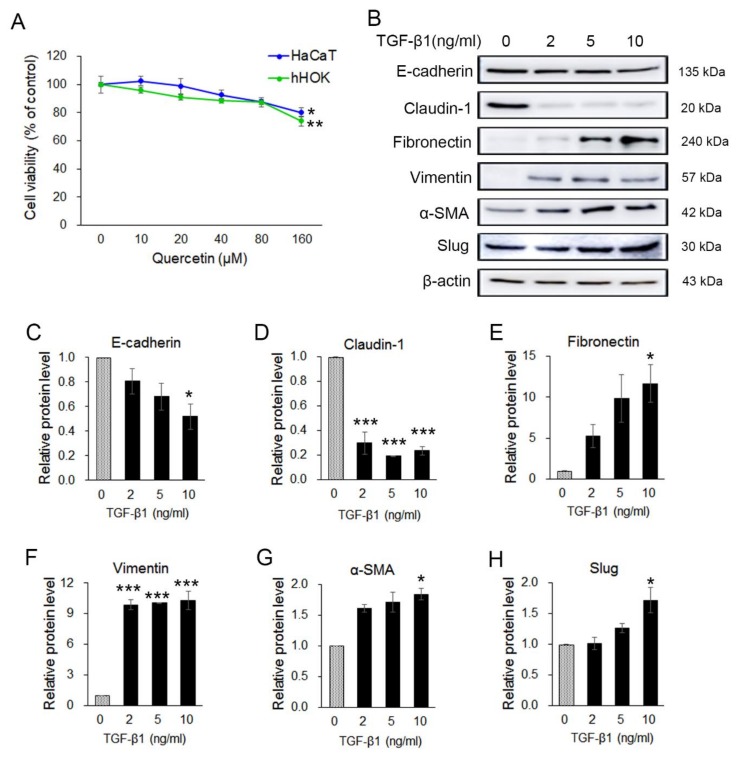
Quercetin was not toxic in normal keratocyte and transforming growth factor β1 (TGF-β1) stimulated EMT in HaCaT cells. (**A**) To examine the effect of quercetin on cell viability, an MTT assay was conducted on HaCaT and nHOK cells. Data are means ± SEM. * *p* < 0.05 vs. the corresponding control (quercetin 0 μM). (**B**) Expression of EMT-related markers such as in TGF-β1-treated HaCaT cells were analyzed by Western blotting. (**C**–**H**) Quantitation of B. The band intensities of each target protein were measured using an image analyzer and presented as relative ratio. Data are means ± SEM. * *p* < 0.05 and *** *p* < 0.005 vs. corresponding control (TGF-β1 0 ng/mL).

**Figure 6 molecules-25-00757-f006:**
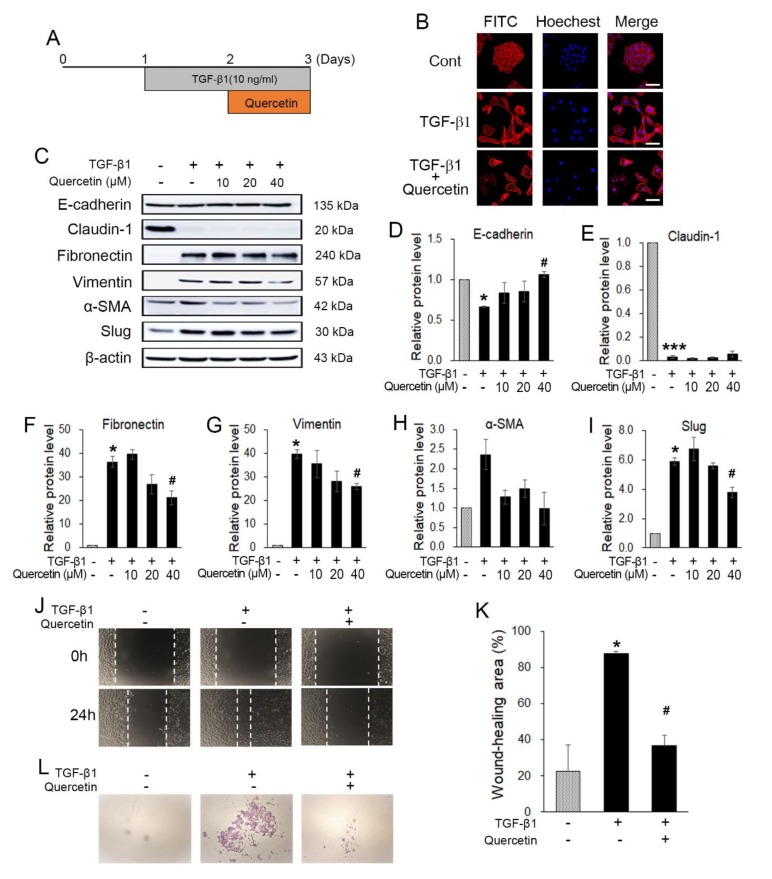
Quercetin inhibited TGF-β1-induced EMT. (**A**) Experimental setup. (**B**) Western blot results showed that quercetin also regulated TGF-β1-induced EMT markers. (**C**) Treatment of TGF-β1 also changed the morphology of HaCaT. Further, quercetin induced morphological recovery. Scale bars: 50 μM. (**D**–**I**) Quantification of hRPTECs viability by MTT assay. (**J**,**K**) A wound-healing assay was conducted to evaluate the TGF-β1-induced EMT migration ability; quercetin attenuated EMT-induced migration in HaCaT cells. (**L**) Quercetin inhibited the invasion capacity of EMT-induced HaCaT cells. Data are the means ± SEM. * *p* < 0.05, and *** *p* < 0.001 vs. the corresponding control (without TGF-β1 and quercetin); ^#^
*p* < 0.05 vs. the corresponding control (with TGF-β1 and without quercetin).

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
