# Peer review of "Quercetin Inhibits Cell Survival and Metastatic Ability via the EMT-Mediated Pathway in Oral Squamous Cell Carcinoma"

_molecules, 2020, doi:10.3390/molecules25030757_

Round 1

Reviewer 1 Report

The results are novel and manuscript is interesting. However, major revision is required.

Specific comments:

1.Please note that HaCaT cells are not normal keratinocytes. They are immortalized cells. Thus, I recommend to add data on the effects of quercetin on normal human keratinocytes with limited proliferation activity (Hayflick limit). This is essential.

2. Please provide quantitative data for all WB analysis. Please also provide information on antibody concentrations and catalog numbers.

3. Please provide SD or SEM for cell cycle data and add statistical analysis.

4.As the authors used very high concentrations of quercetin, please add a paragraph (discussion) on quercetin bioavailability in biological systems.

Author Response

Reviewer 1
Comments to the Author

Specific comments:

Point 1: Please note that HaCaT cells are not normal keratinocytes. They are immortalized cells. Thus, I recommend to add data on the effects of quercetin on normal human keratinocytes with limited proliferation activity (Hayflick limit). This is essential.

Response 1: To examine the viability effect of quercetin on human normal keratinocyte cells, we used nHOK cells (primary normal human oral keratinocytes). The nHOK cells were cultured in serum-free keratinocyte growth basal medium containing the growth factors required for growth of keratinocytes, and growth was very slow. The nHOK cells were treated with various concentrations (0~160 µM) of quercetin for 24 h, and cell viability was determined by an MTT assay. Similar to HaCaT cells, the cell viability of nHOK cells was not changed by the low concentration (10–80 μM) of quercetin compared to the control group but was significantly reduced in 160 μM quercetin treatment (Abstract, Line 148-154, Figure 5A). And we tried to induce EMT by TGF-β1 treatment in nHOK cells but did not go well.

HaCaT cells are immortal keratinocytes, as reviewer points out. Nevertheless, HaCaT cells are widely used in scientific research in vitro because they differentiate into epidermal tissue similar to normal keratinocytes despite the unlimited growth potential. Because the primary normal human oral keratinocytes (nHOK cells) is tricky to EMT stimulation, we used HaCaT cells to investigate whether quercetin also inhibits TGF-β1-induced EMT in human keratinocyte. 

Reference

Boukamp P.; Petrussevska R.T.; Breitkreutz D.; Hornung J.; Markham A.; Fusenig N.E. Normal keratinization in a spontaneously immortalized aneuploid human keratinocyte cell line. J Cell Biol. 1988, 106, 761-771.

Point 2: Please provide quantitative data for all WB analysis. Please also provide information on antibody concentrations and catalog numbers. 

Response 2: We provide quantitative data (Figure 3B, 4B, 5C-H, and 6D-I) and antibodies information (Materials and Methods section) for all western blot analysis according to the reviewer's comment (Line 316-321).

Point 3: Please provide SD or SEM for cell cycle data and add statistical analysis. 

Response 3: Statistical analysis and SEM for cell cycle data are provided in the results section (Line 77-82) and Figure 1B.

Point 4: As the authors used very high concentrations of quercetin, please add a paragraph (discussion) on quercetin bioavailability in biological systems.  

Response 4: I have inserted a discussion about quercetin bioavailability and concentrations used in various cancer cells according to the reviewer's comment (Line 208-219).

Our manuscripts has undergone English language editing by MDPI. The text has been checked for correct use of grammar and common technical terms, and edited to a level suitable for reporting research in a scholarly journal. (English Editing Invoice ID: english-15861)

Reviewer 2 Report

Kim et al explored the effect of quercetin on oral squamous cell carcinoma cell lines.

The authors should explain the specificity of the different cell lines employed and the reasons for the choice.

Rationale for the the examined concentrations in different experiments needs to be added.

L64-67: figure 1 contrasts with result description and a dose-dependency does not appear.

The rationale for the experiment with TGF-β1 should be included.

Statistical analysis should include multifactorial anova for specific experiments.

Methods: details for cell seeding should be corrected (i.e L246 and so on)

Author Response

Reviewer 2
Kim et al explored the effect of quercetin on oral squamous cell carcinoma cell lines.

Point 1: The authors should explain the specificity of the different cell lines employed and the reasons for the choice.

Response 1: We provided the information and references about the specificity of the three different cell lines in introduction section (Line 44-47). OSC20, SAS, and NH22 cell used in this study are a type of oral squamous cell carcinoma. OSC20 and SAS cell are derived from the tongue and HN22 cell are derived from the oral cavity. Oral squamous cell carcinoma (OSCC) occurs on the lips, tongue, salivary glands, gums, lower mouth, pharynx, oral surface in the oral cavity [1-6]. In the 'cnacer facts & figures 2018' provided by the american cancer society, oral cavity were the most common site of onset of oral cancer, followed by pharynx and tongues [12]. Therefore, we used oral squamous cell carcinoma NH22 and tongue squamous cell carcinoma OSC20 and SAS to confirm the oral cancer effect of quercetine.

Point 2: Rationale for the the examined concentrations in different experiments needs to be added.

Response 2: The rationale of quercetin concentration used in this study is presented in the discussion section (Line 208-221).

Point 3: L64-67: figure 1 contrasts with result description and a dose-dependency does not appear. 

Response 3: I have corrected to the following sentence: ‘Cell viability in OSC20 and SAS cells except NH22 cells was reduced in a quercetin dose-dependent manner' (Line 70-71).

Point 4: The rationale for the experiment with TGF-β1 should be included. 

Response 4:  The rationale of TGF-β1 used in this study is presented in the discussion section (Line 243-247). It has been found that transforming growth factor-beta 1 (TGF-b1) induces epithelial endothelial metastasis (EMT) in cancer cells, promoting motility and invasiveness. And TGF-b1 treatment reduces the enhanced expression of Snail1 and Twist1 and subsequently the expression of E-cadherin.

Point 5: Statistical analysis should include multifactorial anova for specific experiments.

Response 5: In statistics, one-way analysis of variance (abbreviated one-way ANOVA) is a technique that can be used to compare means of two or more samples. Statistical analyses in this study were performed with one-way analysis of variance (ANOVA) in SPSS Statistics followed by the Bonferroni post-test when three or more experimental groups were compared (Line 360-361).

Point 6: Methods: details for cell seeding should be corrected (i.e L246 and so on) 

Response 6:  I have corrected details for cell seeding in method section (Line 283, 300, 326, 336, and 351).

Our manuscripts has undergone English language editing by MDPI. The text has been checked for correct use of grammar and common technical terms, and edited to a level suitable for reporting research in a scholarly journal. (English Editing Invoice ID: english-15861)

Round 2

Reviewer 1 Report

The manuscript has been improved.

Reviewer 2 Report

Most of the raised concerns have been addressed and the manuscript appears to be improved after revision